# Mechanical Behaviors Research and the Structural Design of a Bipolar Electrostatic Actuation Microbeam Resonator

**DOI:** 10.3390/s19061348

**Published:** 2019-03-18

**Authors:** Jingjing Feng, Cheng Liu, Wei Zhang, Jianxin Han, Shuying Hao

**Affiliations:** 1Tianjin Key Laboratory of the Design and Intelligent Control of the Advanced Mechatronical System, School of Mechanical Engineering, Tianjin University of Technology, Tianjin 300384, China; syhao@tju.edu.cn; 2Beijing Key Laboratory on Nonlinear Vibrations and Strength of Mechanical Structures, Beijing University of Technology, College of Mechanical Engineering, Beijing 100124, China; 3Tianjin Key Laboratory of High Speed Cutting and Precision Machining, School of Mechanical Engineering, Tianjin University of Technology and Education, Tianjin 300222, China; hanjianxin@tju.edu.cn; 4National Demonstration Center for Experimental Mechanical and Electrical Engineering Education, Tianjin University of Technology, Tianjin 300384, China

**Keywords:** parametric equation, finite element methods, Simulink dynamics simulation, softening-hardening transition points

## Abstract

A class of bipolar electrostatically actuated micro-resonators is presented in this paper. Two parametric equations are proposed for changing the microbeam shape of the upper and lower sections. The mechanical properties of a micro-resonator can be enhanced by optimizing the two section parameters. The electrostatic force nonlinearity, neutral surface tension, and neutral surface bending are considered in the model. First, the theoretical results are verified with finite element results from COMSOL Multiphysics simulations. The influence of section variation on the electrostatic force, pull-in behaviors and safe working area of the micro-resonator are studied. Moreover, the impact of residual stress on pull-in voltage is discussed. The multi-scale method (MMS) is used to further study the vibration of the microbeam near equilibrium, and the relationship between the two section parameters of the microbeam under linear vibration was determined. The vibration amplitude and resonance frequency are investigated when the two section parameters satisfy the linear vibration. In order to research dynamic analysis under the case of large amplitude. The Simulink dynamics simulation was used to study the influence of section variation on the response frequency. It is found that electrostatic softening increases as the vibration amplitude increases. If the nonlinearity initially shows hardening behavior, the frequency response will shift from hardening to softening as the amplitude increases. The position of softening-hardening transition point decreases with the increase of residual stress. The relationship between DC voltage, section parameters, and softening-hardening transition points is presented. The accuracy of the results is verified using theoretical, numerical, and finite element methods.

## 1. Introduction

Nowadays, micro electromechanical system (MEMS) devices are attracting attention due to their small size, low weight, and low energy consumption. Among them, microbeams are a popular micro-component and are widely used in MEMS devices, e.g., as a microswitch [1,2], sensor [3,4], or resonator [5,6,7,8,9]. Therefore, designing and optimizing microbeams can improve the mechanical properties of MEMS devices. An electrostatically actuated microbeam resonator is presented in this paper. 

Electrostatic actuation is one of the most common actuation methods in microstructures [10]. However, electrostatic actuation also has strong nonlinear characteristics [11]. The pull-in effect is the most critical unstable behavior [12]. When the actuation voltage exceeds a critical value, the microbeam will quickly come into contact with the plate, which can cause damage to the device in severe cases. Therefore, determining the unstable position of the microbeam, increasing the pull-in voltage, and the pull-in position of the resonator to improve its mechanical properties is a key issue in micro-resonator design. Abdel-Rahman et al. [13] presented a mathematical model for electrostatic actuation of microbeam resonators, which takes into account the neutral plane tension and nonlinear electrostatic force. It was found that the maximum dimensionless deflection reached 0.39 near pull-in, which is greater than the result calculated with the traditional method. Mobki, Harried et al. [14] presented two models describing unipolar plate of an actuated microbeam and an actuated bipolar plate microbeam; they found that the bipolar plate actuation model has a higher pull-in voltage. In addition, when the voltage ratio at both ends is 1, the increased gap distance ratio leads to decreased attraction voltage, and vice versa.

In addition to studying the pull-in behavior, vibration of the micro-resonator should also be analyzed. Nonlinearity can lead to multiple solutions, leading to jump instability. The relationship between the system’s equivalent natural frequency and DC voltage depends on the strong nonlinearity and electrostatic softening. Moreover, the ideal working condition of the micro-resonator is linear vibration, thus the nonlinearity in the design should be eliminated to the greatest possible extent. Li et al. [15] studied the influence of physical parameters, such as microbeam viscoelasticity and DC voltage, on the system vibration. Finally, the parameters were optimized to determine the relationship between the physical parameters under linear vibration. Han et al. [16] used a spring-mass model to study the vibration of clamped-clamped microbeams and obtained a parameter design diagram relating the DC voltage and initial gap distance under linear conditions. The optimal design voltage under linear conditions was derived. In addition, nonlinearity can be utilized; Li et al. [17] used the modal coupling of a clamped-clamped microbeam to adjust the pull-in voltage and resonant frequency. Based on the coupled vibration behavior between the antisymmetric and symmetric modes, Hu et al. [18] presented a theoretical method for suppressing the midpoint displacement and reducing deformation. However, most early studies were based on rectangular microbeams and focused on optimizing their mechanical properties by adjusting their physical parameters.

As research has deepened, many scholars have found that the mechanical properties of a micro-resonator can be improved by adjusting the microbeam geometry when using the same raw materials. Joglekar and Trivedi [19,20] studied a class of clamped-clamped microbeams and proposed a parametric equation that describes smooth changes along the width of the microbeams. The system mechanical properties are improved by optimizing parameters in the relevant equation, and the results from the method were verified in several cases. On the basis of [19,20], Zhang et al. [21] discussed the influence of the optimized shape on the dynamic response of the microbeam. Kuang and Chen [22] optimized the mechanical properties of a micro-resonator by adjusting the microbeam thickness and gap distance. The required operating voltage range increased by a factor 6. Other studies focus on active control of micro-resonators; Alsaleem and Younis [23] introduced time-delay feedback into the dynamic control of electrostatically actuated micro-resonators. The influence of time delay displacement and velocity feedback on the system dynamic characteristics is discussed in terms of numerical and experimental results. Both kinds of control can improve the system dynamic behavior, enhance stability, and prevent pull-in under positive feedback gain. This is a very effective method for optimized resonator design.

So far, research on microbeam optimization primarily focuses on adjusting the microbeam size [24], using time delay feedback for active control [25,26], and optimizing the shape of the microbeam [27]. In this study, a micro-resonator was optimized by adjusting the microbeam shape, and the number of parametric equations was increased compared to models presented by predecessors. Equations are defined for the upper and lower sections of the microbeam such that the thickness changes uniformly along its length. Each equation has an independent parameter that can be used to adjust variations in the microbeam thickness. The resonator is optimized by choosing the appropriate values of two section parameters. The influence of section on the electrostatic force, small amplitude vibration, and dynamic behavior with large amplitude actuation in bipolar plates are investigated theoretically and with finite element simulations.

The structure of this paper is as follows. The mathematical model (partial differential equation) for electrostatic actuation of a microbeam resonator considering neutral stretching and bending is established in Section 2,. Two parametric equations are proposed for adjusting variations in the microbeam thickness. The Galerkin method is used to simplify the original vibration equation into an ordinary differential equation. In Section 3, the finite element method is used to simulate the electrostatic force on the system, and the influence of two section parameters on the electrostatic force, static pull-in behavior, and safe working area are discussed. In Section 4, the MMS is used to determine the response of the system under small amplitude vibrations. The relationship between the two section parameters is obtained by taking linear vibration as an optimization condition. Variation in the system’s equivalent natural frequency and vibration amplitude are studied given this relationship. COMSOL Multiphysics was used to simulate and verify the natural frequency and electrostatic softening of the optimized micro-resonator. In Section 5, the influence of the section parameters on transitions between softening and hardening of the frequency response curve under large amplitude vibration is discussed. Moreover, the relationship between DC voltage, microbeam thickness, and the transition point is presented. Finally, a summary and conclusions are presented in the last section.

## 2. Mathematical Model

### 2.1. Governing Equation

A schematic of an electrically actuated microbeam is shown in Figure 1. The model consists of two fixed plates and a movable microbeam. Microbeams can be divided into six cases corresponding to various parametric equations. Figure 1a shows ordinary rectangular beams. Figure 1b,c shows a microbridge model, where the thickness does not change along beam length. Figure 1d,e shows thickened and thinned beams, respectively. both of which are symmetrical around the neutral plane. Figure 1f shows an irregular beam. A capacitor forms between the microbeam and the plate by applying a DC voltage $V_{dc}$ and an AC voltage $V_{ac}\cos(\Omega t)$ to the plate, thus generating an electrostatic force. 

Changes in the microbeam shape will produce an uneven electrostatic force between the upper and lower parts of the microbeam. The distance between the microbeam section at the fixed end and electrode plate is d, and the microbeam exhibits lateral motion under the action of electrostatic forces. A model whose thickness varies along microbeam length is considered here, where l and b are the length and the width of the microbeam, respectively. h is the thickness of microbeam at the clamped ends. The cross-sectional area and moment of inertia of the two clamped sides are $A_{0}=bh$ and $I_{0}=bh^{3}/12$, respectively. The microbeam thickness is determined by $y_{1}(x)=\frac{h}{2}+\lambda_{1}h\;\sin\frac{\pi x}{l}$ and $y_{2}(x)=-\frac{h}{2}+\lambda_{2}h\;\sin\frac{\pi x}{l}$, where $\lambda_{1}$ and $\lambda_{2}$ are section parameters, and the microbeam section curvature changes with $\lambda_{1}$ and $\lambda_{2}$. The cross-sectional area and moment of inertia are $A(x)=A_{0}(1+(\lambda_{1}-\lambda_{2})\sin(\pi x/l))$ and $I(x)=I_{0}{\left(1+(\lambda_{1}-\lambda_{2})\sin(\pi x/l)\right)}^{3}$, respectively. In order to achieve better mechanical properties when both of microbeams consume same amount of material during their fabrication, it is assumed that the volume of the microbeam is constant. The relationship between h and $\lambda_{1}$, $\lambda_{2}$ is found to be $h=\frac{h_{0}(\pi -2\lambda_{1}+2\lambda_{2})}{\pi }$ by setting the microbeam volume does not change when the shape changes, where $h_{0}$ is the microbeam thickness when the beam is rectangular. The capacitance varies in microbeam different positions due to the changes in the gap between the microbeam and electrode plate. The capacitances of the upper and lower plates are
(1)$$\begin{array}{l} C_{1}=\frac{\varepsilon_{0}\varepsilon_{r}bl}{d-y_{1}(x)+\frac{h}{2}-y(x,t)}\\ C_{2}=\frac{\varepsilon_{0}\varepsilon_{r}bl}{d+y_{2}(x)+\frac{h}{2}+y(x,t)} \end{array}$$
where $\varepsilon_{0}$ is the dielectric constant in free space, $\varepsilon_{r}$ is the relative dielectric permittivity in the gap relative to free space, $C_{1}$ is the capacitance between the upper plate and the microbeam, and $C_{2}$ is the capacitance between the upper plate and the microbeam.

The total energy stored by two capacitors is

(2)$$W=\frac{1}{2}C_{1}{\left(V_{dc}+V_{ac}\right)}^{2}+\frac{1}{2}C_{2}{\left(V_{dc}\right)}^{2}.$$

The electrostatic force is

(3)$$F=\frac{\partial (\frac{1}{2}C_{1}{\left(V_{dc}+V_{ac}\cos(\Omega t)\right)}^{2})}{\partial (d-y_{1}(x)+\frac{h}{2}-y(x,t))}-\frac{\partial (\frac{1}{2}C_{2}{\left(V_{dc}\right)}^{2})}{\partial (d+y_{1}(x)+\frac{h}{2}+y(x,t))}$$

The equation of motion considers changes in the cross-sectional area and the electrostatic force on the bipolar plate, the equation is therefore rewritten as follows [28]:(4)$$\begin{array}{l} \frac{\partial^{2}}{\partial x^{2}}(EI(x)\frac{\partial^{2}y(x,t)}{\partial x^{2}})+\rho A(x)\frac{\partial^{2}y(x,t)}{\partial t^{2}}+c\frac{\partial y(x,t)}{\partial t}=(N\\ +\frac{E}{2l}\int_{0}^{1}A(x)({\left(\frac{\partial y(x,t)}{\partial x}\right)}^{2}+\frac{\partial y(x,t)}{\partial x}\frac{d(y_{1}(x)+y_{2}(x))}{dx})dx)(\frac{\partial^{2}y(x,t)}{\partial x^{2}}+\frac{d^{2}(y_{1}(x)+y_{2}(x))}{2dx^{2}})\\ +\frac{\varepsilon_{0}\varepsilon_{r}b{\left[V_{dc}+V_{ac}\cos(\Omega t)\right]}^{2}}{2{\left(d-y_{1}(x)+\frac{h}{2}-y(x,t)\right)}^{2}}-\frac{\varepsilon_{0}\varepsilon_{r}bV_{dc}{}^{2}}{2{\left(d+y_{2}(x)+\frac{h}{2}+y(x,t)\right)}^{2}} \end{array}$$

The boundary conditions are
(5)$$y(0,t)=0,\;\frac{\partial y}{\partial x}(0,t)=0,\;y(l,t)=0,\;\frac{\partial y}{\partial x}(l,t)=0,$$
where c is the damping coefficient, and E and ρ are the elastic modulus and density, respectively.

In Equation (4), N is the residual stress, the integral term is the tension and deformation of neutral plane.

For the convenience of calculation, consider following dimensionless:(6)$$\overset{⌢}{x}=\frac{x}{l},\overset{⌢}{y}=\frac{y}{d},{\overset{⌢}{y}}_{1}=\frac{y_{1}}{d},{\overset{⌢}{y}}_{2}=\frac{y_{2}}{d},\overset{⌢}{A}(\overset{⌢}{x})=\frac{A(x)}{A_{0}},\overset{⌢}{I}(\overset{⌢}{x})=\frac{I(x)}{I_{0}},\overset{⌢}{t}=\frac{t}{T},\omega =\frac{\Omega t}{\overset{⌢}{t}}.$$

Substituting Equation (6) into (4) and (5) yields the dimensionless bending vibration equation:(7)$$\begin{array}{l} \frac{\partial^{2}}{\partial {\overset{⌢}{x}}^{2}}(I(\overset{⌢}{x})\frac{\partial^{2}\overset{⌢}{y}(x,t)}{\partial {\overset{⌢}{x}}^{2}})+A(\overset{⌢}{x})\frac{\partial^{2}\overset{⌢}{y}(x,t)}{\partial {\overset{⌢}{t}}^{2}}+c^{\prime }\frac{\partial \overset{⌢}{y}(x,t)}{\partial \overset{⌢}{t}}-(\overset{⌢}{N}\\ +\alpha_{2}\int_{0}^{1}A(\overset{⌢}{x})({\left(\frac{\partial \overset{⌢}{y}(x,t)}{\partial \overset{⌢}{x}}\right)}^{2}+\frac{\partial \overset{⌢}{y}(x,t)}{\partial \overset{⌢}{x}}\frac{d({\overset{⌢}{y}}_{1}(x)+{\overset{⌢}{y}}_{2}(x))}{d\overset{⌢}{x}})d\overset{⌢}{x})(\frac{\partial^{2}\overset{⌢}{y}(x,t)}{\partial {\overset{⌢}{x}}^{2}}\\ +\frac{d^{2}({\overset{⌢}{y}}_{1}(x)+{\overset{⌢}{y}}_{2}(x))}{2d{\overset{⌢}{x}}^{2}})=\alpha_{1}(\frac{1}{{\left(1-{\overset{⌢}{y}}_{1}(x)+\frac{h}{2d}-\overset{⌢}{y}(x,t)\right)}^{2}}-\frac{1}{{\left(1+{\overset{⌢}{y}}_{2}(x)+\frac{h}{2d}+\overset{⌢}{y}(x,t)\right)}^{2}})\\ +\frac{2\alpha_{1}\rho \cos(\omega \overset{⌢}{t})}{{\left(1-{\overset{⌢}{y}}_{1}(x)+\frac{h}{2d}-\overset{⌢}{y}(x,t)\right)}^{2}}+\frac{\alpha_{1}\rho^{2}\cos^{2}(\omega \overset{⌢}{t})}{2{\left(1-{\overset{⌢}{y}}_{1}(x)+\frac{h}{2d}-\overset{⌢}{y}(x,t)\right)}^{2}} \end{array}$$

The corresponding boundary condition becomes
(8)$$\overset{⌢}{y}(0,\overset{⌢}{t})=0,\frac{\partial \overset{⌢}{y}}{\partial \overset{⌢}{x}}(0,\overset{⌢}{t})=0,\overset{⌢}{y}(1,\overset{⌢}{t})=0,\frac{\partial \overset{⌢}{y}}{\partial \overset{⌢}{x}}(1,\overset{⌢}{t})=0,$$
where
$$T=\sqrt{\frac{l^{4}\rho A_{0}}{EI_{0}}};c^{\prime }=\frac{cl^{4}}{EI_{0}T};\alpha_{1}=\frac{\varepsilon_{0}\varepsilon_{r}bl^{4}Vdc^{2}}{2EI_{0}d^{3}};\rho =\frac{V_{ac}}{V_{dc}},\alpha_{2}=6{\left(\frac{d}{h}\right)}^{2}.$$

In the following figure, the “$\begin{array}{c} \overset{⌢}{} \end{array}$” symbol has been removed from the dimensionless parameters for convenience.

The fringing fields of electrostatic force are not taken into account in the above calculation. In order to observe the accuracy of electrostatic force, the electrostatic force in Equation (7) is compared with the electrostatic force considering edge effect. The results are shown in Figure 2. It can be found that the two results are consistent. Therefore, the infinite plate model is applied for convenience.

### 2.2. Galerkin Expansion

The deflection of the microbeam is defined as follows:(9)$$y(x,t)=\sum_{i=1}^{\infty }u_{i}(t)\phi_{i}(x).$$

The boundary conditions are

(10)$$\phi_{i}(0)=\phi_{i}(1)=\phi_{i}{}^{\prime }(0)=\phi_{i}{}^{\prime }(1)=0,$$

$u_{i}(t)$ is the *i*-th modal coordinate amplitude, and $\phi_{i}(x)$ is the *i*-th order undamped linear orthogonal mode function. According to results from the literature [29], a model with a single degree-of-freedom can sufficiently capture all key nonlinear aspects in the Galerkin approximation when studying the primary resonance. Assuming y(x,t)=u(t)ϕ(x), the existing conventional calculation is usually based on a Taylor expansion of the electrostatic force or by multiplying the denominator term of the electrostatic force [30]. The first method is a simple calculation, but the higher order displacement terms will be eliminated after Taylor expansion, which decreases the accuracy of the results. In order to accurately describe the electrostatic force while retaining the nonlinear characteristics of the electrostatic force as much as possible, a second calculation scheme is adopted in this paper. The denominator term is eliminated by multiplying ${\left(1-y_{1}(x)+\frac{h}{2d}-y(x,t)\right)}^{2}{\left(1+y_{2}(x)+\frac{h}{2d}+y(x,t)\right)}^{2}$ at vibration equation. Since $V_{ac}$ is much smaller than $V_{dc}$ in the micro-resonator, the following calculation omits the $\rho^{2}$ term. Substituting Equation (9) into (7), multiplying by ϕ(x), and integrating from x = 0 to 1 yields the following equation:(11)$$g\ddot{u}+\mu \dot{u}+k_{0}+k_{1}u+k_{2}u^{2}+k_{3}u^{3}+k_{4}u^{4}+k_{5}u^{5}+k_{6}u^{6}+k_{7}u^{7}=2\alpha_{1}\rho (v_{ac1}+v_{ac2}u+v_{ac2}u^{2})\cos(\omega t),$$
where, $\dot{u}=du/dt$. Expressions for g, μ, $k_{1}$, $k_{2}$, $k_{3}$, $k_{4}$, $k_{5}$, $k_{6}$, $k_{7}$, $v_{dc1}$, $v_{dc2}$, $v_{ac1}$, $v_{ac2}$, and $v_{ac3}$ are shown in Appendix A.

The static equilibrium equation can be written as follows:(12)$$k_{0}+k_{1}u+k_{2}u^{2}+k_{3}u^{3}+k_{4}u^{4}+k_{5}u^{5}+k_{6}u^{6}+k_{7}u^{7}=0.$$

Without considering damping and AC disturbances, the following Hamiltonian system corresponding to the dimensionless dynamic equation can be obtained:(13)$$\left\{ \begin{array}{l} \dot{u}=v\\ \dot{v}=-\frac{1}{g}(k_{0}+k_{1}u+k_{2}u^{2}+k_{3}u^{3}+k_{4}u^{4}+k_{5}u^{5}+k_{6}u^{6}+k_{7}u^{7}) \end{array}\right..$$

The potential energy function is
(14)$$V(\xi )=\int_{0}^{\xi }\dot{v}du,$$
and the corresponding Hamiltonian is

(15)$$H(u,v)=V(u)+\frac{1}{2}v^{2}.$$

One should note here that the maximum lateral displacement of the microbeam is at the midpoint viz., $y_{\max}=\phi (0.5)u\in [-1-\lambda_{2}\delta ,1-\lambda_{1}\delta ]$. The value of the modal function is ϕ(0.5)=1.59, thus the range of u is $u\in [\frac{-1-\lambda_{2}\delta }{1.59},\frac{1-\lambda_{1}\delta }{1.59}]$.

## 3. Static Analysis

First, a static analysis of the system is conducted. The safe region of the micro-resonator can be obtained from static analysis. Only by guaranteeing that the device can operate safely under the selected parameters can the research presented here be meaningful. Here, the accuracy of the electrostatic force obtained from theoretical analysis is verified. All possible microbeam cases are selected. The capacitance the between plates and microbeam was simulated using COMSOL Multiphysics, and the simulation model is shown in Figure 3. A voltage was applied at d on the clamped end of the microbeam while the microbeam was grounded. The entire model was placed in air. Assuming L=400 μm, b=45 μm, d=3 μm, $\rho =2.33\times 10^{3}\;\mathrm{kg}/m^{3}$, E=165 Gpa, and $\varepsilon_{0}=8.85\times 10^{-12}\;F/m$, and the thickness of the clamped-clamped ends is h=2 μm. The entire simulation was performed using the steady state solver.

The results from two methods are compared in Figure 4. This comparison shows that the two results are consistent. Although there is a slight deviation at larger voltage, the effect is insignificant. At this time, the voltage may have exceeded the pull-in voltage. Thus, high voltage cannot be used during operation. The specific pull-in voltage will be calculated later. Overall, the simulation results verify the correctness of theory.

The theoretical results shown in Figure 4 are used to determine the influence of section variation on the electrostatic forces. The middle thickness of microbeam becomes thinner and the thickness of clamped-clamped ends becomes thicker when $\lambda_{1}$ is held constant and $\lambda_{2}$ decreases. The system potential can be reduced by decreasing $\lambda_{2}$, as shown in Figure 5. This indicates that the potential energy is larger when the microbeam is thinner at the midpoint for a given voltage due to the greater electrostatic force.

Next, the influence of the section parameters on static pull-in behaviors will be studied. In order to facilitate observation, the special case of a rectangular microbeam is added as a comparison, where the pull-in behavior is indicated with the black curve in Figure 6a,b. The physical parameters of the microbeam are L=400 μm, b=45 μm, d=3 μm, $h_{0}=2\;\mu m$, $E=1.65\times 10^{11}\;\mathrm{Pa}$, and $\rho =2.33\times 10^{3}\;\mathrm{kg}/m^{3}$. Figure 6a,b shows the case where the upper section parameters are $\lambda_{1}=0.2$ and $\lambda_{1}=-0.2$. The lower section moves gradually from −0.3 to 0.3. At this time, the middle of the microbeam becomes thinner and the ends become thicker. The red curve corresponds to $\lambda_{1}=0.2$ and $\lambda_{2}=-0.2$, while the green curve corresponds to $\lambda_{1}=-0.2$ and $\lambda_{2}=0.2$. The microbeam is symmetric in the neutral plane in these two cases. Figure 6 shows that the static pull-in position gradually increases as the microbeam becomes thinner, eventually leading to secondary pull-in. Figure 6b shows the case where $\lambda_{2}=0.2$ and $\lambda_{2}=0.3$. Asymmetry of the microbeam will cause the static equilibrium point shift to the side with larger $\lambda_{1}$ and $-\lambda_{2}$. Greater asymmetry leads to a larger migration distance. When $\lambda_{1}=-\lambda_{2}$, the microbeam is symmetric around the neutral plane and the equilibrium point position is symmetric around the x-axis. If $\lambda_{1}=-\lambda_{2}>0$, the midpoint of the microbeam is thicker than at the ends, and vice versa if $\lambda_{1}=-\lambda_{2}<0$. Comparing the three curves of red, black, and green in Figure 6a,b shows that the range of pull-in voltage will increase and secondary pull-in phenomenon will be more likely when the midpoint is thinner than the ends.

In addition, static displacement jump will occur in the asymmetric microbeam. One can see from the case where $\lambda_{2}=0.3$ in Figure 6b that the static displacement jumps from positive to negative when the DC voltage amplitude increases to $V_{dc}=87\;V$. A comparison of Figure 6a,b shows that the static pull-in voltage mainly depends on the larger side of $\lambda_{1}$ and $-\lambda_{2}$. As shown in Figure 6a, the static pull-in position of $\lambda_{2}$ from −0.3 to −0.2 varies greatly when $\lambda_{1}=0.2$, but the increasing trend in the static pull-in position decreases significantly when $\lambda_{2}$ ranges from −0.2 to 0.3. This arises because the static suction position at this time is mainly determined by $\lambda_{1}$. 

In order to discuss this phenomenon more intuitively, Assume $V_{dc}=82.5V$ and draw the potential energy of $\lambda_{1}=-0.2$ and $\lambda_{2}=0.2$ to observe the influence of another section variation on system safe area. When $\lambda_{1}=-0.2$, the pull-in points of $\lambda_{2}=0.19$, $\lambda_{2}=0.2$, and $\lambda_{2}=0.21$ correspond to green, red, and blue points in Figure 7a, respectively. The lowest barrier energy is significantly lower when compared to the case when $\lambda_{2}=0.2$, $\lambda_{2}=0.21$. Similarly, in Figure 7b, when $\lambda_{2}=0.2$, the phenomenon of $\lambda_{1}=-0.21$, $\lambda_{1}=-0.2$ and $\lambda_{1}=-0.19$ is same as that of Figure 7a. The static pull-in position primarily depends on the larger of $\lambda_{1}$ and $-\lambda_{2}$. Meanwhile, Figure 7a,b is symmetric around the y-axis, indicating that the corresponding pull-in voltages are equal and the pull-in positions are opposite when the section parameters of two cases are opposite to each other.

Therefore, reducing $\lambda_{1}$ or increasing $\lambda_{2}$ in the design can improve the working range and stability of the resonator. On the other hand, static pull-in voltage primarily depends on the larger of $\lambda_{1}$ and $-\lambda_{2}$. When one section of the microbeam is determined, the two fixed ends become thicker and more energy is lost at the ends when the other section is closer to the neutral plane, but the position of pull-in voltage does not change much. Therefore, taking $\lambda_{1}=-\lambda_{2}$ can increase the pull-in voltage.

Residual stress may occur in the process of fabrication or operation of devices and the mechanical properties will be influenced. In order to accurately predict the safe working area of the device, the influence of residual stress on pull-in voltage and secondary pull-in will be discussed below. As shown in Figure 8, the pull-in voltage increases with the increase of residual stress. Figure 8a shows that when $\lambda_{1}=0.2$, the pull-in voltage increases with the increase of $\lambda_{2}$ and before $\lambda_{2}<-0.2$, the pull-in voltage increases faster than after $\lambda_{2}>-0.2$. This proves once again that the static pull-in voltage primarily depends on the larger of $\lambda_{1}$ and $-\lambda_{2}$. Figure 8b shows the case of $\lambda_{2}=0.2$, It can be seen that the pull-in voltage decreases as $\lambda_{1}$ increases. Secondary pull-in phenonemon occurs in yellow and blue regions. Each $\lambda_{2}$ corresponds to three voltage values in this region. The dotted line indicates that the voltage on each line corresponds to the position in the static displacement picture. When $\lambda_{1}<-0.2$, the secondary pull-in region is close to the lower plate and when $\lambda_{1}>-0.2$, the secondary pull-in region is close to upper plate. In addition, the larger the yellow and blue areas, the larger the secondary pull-in range is. Therefore, the increase of residual stress will reduce the range of secondary pull-in area.

## 4. Dynamic Analysis

It is very important to conduct dynamic analysis of MEMS devices. The influence of section variation on the frequency response and bifurcation behavior can be observed through dynamic analysis, so as to optimize micro-resonator. MMS is used in the following to calculate small amplitude vibrations of the micro-resonator in the stable region. Here, we define $u=u_{D}+u_{A}$, where $u_{D}$ is the response under DC voltage and $u_{A}$ is the response under AC voltage.

First, $u_{D}$ can be obtained from Equation (12). Substituting $u=u_{D}+u_{A}$ into Equation (11), ignoring the nonlinear terms above third order and nonlinear damping in Equation (11), and eliminating the term corresponding to the equilibrium position yields the small vibration equation at the equilibrium position:(16)$${\ddot{u}}_{A}+c*{\dot{u}}_{A}+\omega_{n}^{2}u_{A}+a_{q}u_{A}{}^{2}+a_{c}u_{A}{}^{3}+a_{m}u_{A}{\ddot{u}}_{A}+a_{n}u_{A}{}^{2}{\ddot{u}}_{A}=f\cos(\omega t).$$

The meaning of each coefficient is shown in Appendix B.

Because $V_{ac}$ is much smaller than $V_{dc}$ in the micro-resonator, we consider the cases $V_{dc}=O(1)$ and $V_{ac}=O(\varepsilon^{3})$, where ε is a small dimensionless parameter. Equation (16) can be rewritten as

(17)$${\ddot{u}}_{A}+\varepsilon^{2}c*{\dot{u}}_{A}+\omega_{n}^{2}u_{A}+a_{q}u_{A}{}^{2}+a_{c}u_{A}{}^{3}+a_{m}u_{A}{\ddot{u}}_{A}+a_{}u_{A}{}^{2}{\ddot{u}}_{A}=\varepsilon^{3}f\cos(\omega t).$$

In order to describe the primary resonance accurately, a tuning parameter σ is introduced and defined as

(18)$$\omega =\omega_{n}+\varepsilon^{2}\sigma .$$

The approximate solution to Equation (17) is expressed as follows:(19)$$u_{A}(t,\varepsilon )=\varepsilon u_{A1}(T_{0},T_{1},T_{2})+\varepsilon^{2}u_{A2}(T_{0},T_{1},T_{2})+\varepsilon^{3}u_{A3}(T_{0},T_{1},T_{2}),$$
where $T_{n}=\varepsilon^{n}t,n=(0,\;1,\;2)$.

Substituting Equations (18) and (19) into (17) and gathering like powers of ε yields:(20)$$O(\varepsilon^{1}):D_{0}^{2}u_{A1}+\omega_{n}^{2}u_{A1}=0,$$
(21)$$O(\varepsilon^{2}):D_{0}^{2}u_{A2}+\omega_{n}^{2}u_{A2}=-2D_{0}D_{1}u_{A1}-a_{c}u_{A1}^{2}-a_{m}D_{0}^{2}u_{A1},$$
(22)$$\begin{array}{ll} O(\varepsilon^{3}): & D_{0}^{2}u_{3}+\omega_{n}^{2}u_{3}=-2D_{0}D_{1}u_{A2}-2D_{0}D_{2}u_{A1}-D_{1}^{2}u_{A1}-c^{*}D_{0}u_{A1}-2a_{m}u_{A1}D_{0}D_{1}u_{A1}\\ & -a_{n}u_{A1}D_{0}^{2}u_{A2}-a_{m}u_{A2}D_{0}^{2}u_{A1}-a_{n}u_{{}_{A1}}^{2}D_{0}^{2}u_{A1}-2a_{c}u_{1}u_{2}-a_{c}u_{1}^{3}+f\cos(\omega_{n}T_{0}+\sigma T_{2}) \end{array}$$
where $D_{n}=\frac{\partial }{\partial T_{n}},n=(0,\;1,\;2)$.

The general solution to Equation (20) can be written as

(23)$$u_{A1}(T_{0},T_{1},T_{2})=A(T_{1},T_{2})e^{i\omega_{n}T_{0}}+\bar{A}(T_{1},T_{2})e^{-i\omega_{n}T_{0}}.$$

Substituting Equation (23) into (21) yields:(24)$$D_{0}^{2}u_{A2}+\omega_{n}^{2}u_{A2}=-2i\omega_{n}\frac{\partial A}{\partial T_{1}}e^{i\omega_{n}T_{0}}-a_{q}(A^{2}e^{2i\omega_{n}T_{0}}+A\bar{A})+a_{m}\omega_{n}^{2}(A^{2}e^{2i\omega_{n}T_{0}}+A\bar{A})+cc,$$
where cc denotes the complex conjugate.

To eliminate the secular term, one needs
(25)$$-2i\omega_{n}\frac{\partial A}{\partial T_{1}}e^{i\omega_{n}T_{0}}=0,$$
which indicates that A is only a function of $T_{2}$. Thus, Equation (24) becomes

(26)$$D_{0}^{2}u_{A2}+\omega_{n}^{2}u_{A2}=(a_{m}\omega_{n}^{2}-a_{q})(A^{2}e^{2i\omega_{n}T_{0}}+A\bar{A})+cc.$$

The solution $u_{A2}$ can be written as follows:(27)$$u_{A2}(T_{0},T_{2})=\frac{(a_{q}-a_{m}\omega_{n}^{2})A^{2}}{3\omega_{n}^{2}}e^{2i\omega_{n}T_{0}}-\frac{(a_{q}-a_{m}\omega_{n}^{2})A\bar{A}}{\omega_{n}^{2}}+cc.$$

Substituting Equations (23) and (27) into Equation (22) yields the secular terms

(28)$$2i\omega_{n}\frac{\partial A}{\partial T_{1}}+c*i\omega A-\frac{(10a_{q}-a_{m}\omega_{n}{}^{2})(a_{q}-a_{m}\omega_{n}{}^{2})A^{2}\bar{A}}{3\omega_{n}{}^{2}}+3(a_{c}-a_{n}\omega_{n}{}^{2})A^{2}\bar{A}-\frac{f}{2}e^{i\sigma T_{2}}=0.$$

At this point, it is convenient to express A in polar form:(29)$$A=\frac{1}{2}a(T_{2})e^{i\beta (T_{2})}+cc.$$

Substituting Equation (29) into Equation (28) and separating the imaginary and real parts yields:(30)$$\frac{Da}{DT_{2}}=-\frac{c*}{2}a-\frac{f}{2\omega_{n}}\sin\varphi ,$$
(31)$$a\frac{D\varphi }{DT_{2}}=\sigma a+a^{3}(\frac{\left(10a_{q}-a_{m}\omega_{n}^{2})(a_{q}-a_{m}\omega_{n}^{2}\right)}{24\omega_{n}^{3}}-\frac{3(a_{c}-a_{n}\omega_{n}^{2})}{8\omega_{n}})+\frac{f}{2\omega_{n}}\cos\varphi ,$$
where $\varphi =\sigma T_{2}-\beta$.

The steady-state response can be obtained by imposing the condition $\frac{Da}{DT_{2}}=\frac{D\varphi }{DT_{2}}=0$. Finally, the frequency response equation can be derived as follows:(32)$$a^{2}({\left(\frac{c*}{2}\right)}^{2}+{\left(\sigma +a^{2}\xi \right)}^{2})={\left(\frac{f}{2\omega_{n}}\right)}^{2},$$
where $\xi =\frac{\left(10a_{q}-a_{m}\omega_{n}^{2})(a_{q}-a_{m}\omega_{n}^{2}\right)}{24\omega_{n}^{3}}-\frac{3(a_{c}-a_{n}\omega_{n}^{2})}{8\omega_{n}}$.

The peak vibration amplitude and backbone curve are $a_{\max}=f/(\mu \omega_{n})$ and $\omega =\omega_{n}-\kappa a_{\max}$, respectively. The stability of the periodic solution can be determined using a method found in the literature [29,31].

### 4.1. Parameter Optimization

In the previous section, the stable region of the micro-resonator is optimized by static analysis, followed by conducting dynamic analysis. Previous research [29] has shown that the section parameter λ changes the softening and hardening behavior of the frequency response. It is noteworthy that there will be a moment of linear vibration in the transformation between softening and hardening, which is an ideal condition for micro-resonators. Therefore, the MEMS resonator is designed based on this condition. Assume d=2 μm. First, one finds by calculation that the microbeam vibration is linear and rectangular when $V_{dc}=23.95\;V$. The sectional parameters $\lambda_{1}$ and $\lambda_{2}$ under linear vibration are shown in Figure 9.

The coordinates of these special points listed in Table 1 are shown in Figure 9. The frequency response in nine cases was studied using MMS. In order to verify the above theoretical results, the long-time integral method (LTI) is used to calculate Equation (11) and the numerical solution is obtained. First, the frequency responses of D, H, and I are shown in Figure 10. Adjusting the AC voltage amplitude to a frequency response maximum amplitude of 0.1 shows that the vibration inside the curve exhibits hardening while the outer side exhibits softening. The point D corresponds to linear vibration in the special case of $\lambda_{1}=0$ and $\lambda_{2}=0$.

Next, points A, B, C, and F in Figure 9 are analyzed to determine the influence of section variation on static pull-in so as to increase the working range of the micro-resonator, where point B and F are symmetric about y=x. Figure 11a shows that the maximum static suction voltage is obtained when $\lambda_{1}=-0.18$ and $\lambda_{2}=0.55$. This also verifies the conclusion in Section 3, namely that the pull-in voltage is determined by the larger value out of $\lambda_{1}$ and $-\lambda_{2}$. The static pull-in curve is symmetric around x=0 when the point in Figure 7 is symmetric about y=x, where the static pull-in voltage is equal and pull-in position is the opposite, as shown in Figure 11b.

The equivalent frequency and amplitude of the microbeam in different conditions are also important for studying the micro-resonator. Figure 12 shows the frequency response curves in different conditions. It is found that the equivalent frequencies and amplitudes of various points in Figure 9 are different. In addition, the points that are symmetric about y=x in Figure 12 have the same equivalent frequency and different amplitudes. The vibration amplitude in region I is relatively large.

The variation in the resonance frequency and amplitude with the sectional parameters $\lambda_{1}$ and $\lambda_{2}$ are studied based on the results in Figure 9. Figure 13a shows that the resonance frequency is largest at point A, and the resonance frequency first decreases and then increases from point A to point B. From point B to point D, the resonance frequency decreases rapidly, reaching point $\lambda_{1}=0$, $\lambda_{2}=0$, and the resonance frequency is smallest. The trends of G, E, and F are the same as those of A, B, and C, and the corresponding equivalent frequency is equal at the point where y=x is symmetric. As shown in Figure 13b, the resonance amplitude increases from point A to point C. For the point corresponding to the y = x symmetry in Figure 9, the amplitude is larger when $\lambda_{1}$ is larger. The maximum amplitude appears between C and D.

### 4.2. Dynamic Analysis with Large Amplitude

Since traditional MMS is only suitable for small vibrations, the traditional MMS will produce incorrect results when vibration amplitude exceeds a certain value [32]. In order to study the mechanical properties of the system when the vibration exceeds a certain amplitude, a new multi-scale method (NMMS) is introduced by combining the homotropy idea with the multi-scale method [33]. Equation (11) is solved using a Simulink dynamics simulation in order to verify the accuracy of the results.

It is assumed that the final vibration frequency is equal to the excitation frequency. A scaling parameters ε can be used to convert Equation (11) into the following form

(33)$$\begin{array}{ll} \ddot{u}+\omega^{2}u= & \varepsilon \{-\frac{c^{\prime }C_{22}}{B_{22}}\dot{u}+(\omega^{2}-\frac{k_{1}}{B_{22}})u-\frac{k_{0}}{B_{22}}-\frac{k_{2}}{B_{22}}u^{2}-\frac{k_{3}}{B_{22}}u^{3}-\frac{k_{4}}{B_{22}}u^{4}-\frac{k_{5}}{B_{22}}u^{5}-\frac{k_{6}}{B_{22}}u^{6}-\frac{k_{7}}{B_{22}}u^{7}\\ & +\frac{2B_{12}-2B_{21}}{B_{22}}u\ddot{u}+\frac{4B_{11}-B_{20}-B_{02}}{B_{22}}u^{2}\ddot{u}+\frac{2B_{10}-2B_{01}}{B_{22}}u^{3}\ddot{u}-\frac{B_{00}}{B_{22}}u^{4}\ddot{u}\\ & +\frac{2\alpha_{1}\rho }{B_{22}}(v_{ac1}+v_{ac2}u+v_{ac2}u^{2})\cos(\omega t)\} \end{array}$$

The approximate solution to Equation (33) is expressed as follows:(34)$$u(t,\varepsilon )=u_{0}(T_{0},T_{1},T_{2})+\varepsilon u_{1}(T_{0},T_{1},T_{2})+\varepsilon^{2}u_{2}(T_{0},T_{1},T_{2})+\cdots .$$

Substituting Equation (34) into (33) and equating coefficients of like powers of ε yields the following equations:(35)$$O(\varepsilon^{0}):D_{0}^{2}u_{0}+\omega^{2}u_{0}=0.$$
(36)$$\begin{array}{ll} O(\varepsilon^{1}): & D_{0}^{2}u_{1}+\omega^{2}u_{1}=-2D_{0}D_{1}u_{1}-\frac{c^{\prime }C_{22}}{B_{22}}D_{0}u+(\omega^{2}-\frac{k_{1}}{B_{22}})u_{0}-\frac{k_{0}}{B_{22}}-\frac{k_{2}}{B_{22}}u_{0}{}^{2}-\frac{k_{3}}{B_{22}}u_{0}{}^{3}\\ & -\frac{k_{4}}{B_{22}}u_{0}{}^{4}-\frac{k_{5}}{B_{22}}u_{0}{}^{5}-\frac{k_{6}}{B_{22}}u_{0}{}^{6}-\frac{k_{7}}{B_{22}}u_{0}{}^{7}+\frac{2B_{12}-2B_{21}}{B_{22}}u_{0}D_{0}^{2}u_{0}+\frac{4B_{11}-B_{20}-B_{02}}{B_{22}}u_{0}{}^{2}D_{0}^{2}u_{0}\\ & +\frac{2B_{10}-2B_{01}}{B_{22}}u_{0}{}^{3}D_{0}^{2}u_{0}-\frac{B_{00}}{B_{22}}u_{0}{}^{4}D_{0}^{2}u_{0}+\frac{2\alpha_{1}\rho }{B_{22}}(v_{ac1}+v_{ac2}u_{0}+v_{ac2}u_{0}{}^{2})\cos(\omega t) \end{array}$$
where: $D_{n}=\frac{\partial }{\partial T_{n}}$, n=(0, 1, 2).

The general solution to Equation (35) can be written in the following form:(37)$$u_{0}=A(T_{1})\cos(\omega T_{0}+\beta (T_{1})),$$
where $A(T_{1})$ is vibration amplitude and $\beta (T_{1})$ is the vibration phase.

The secular term derived from MMS is expressed as:(38)$$\begin{array}{l} \int_{0}^{2\pi }(-2D_{0}D_{1}u_{1}-\frac{c^{\prime }C_{22}}{B_{22}}D_{0}u+(\omega^{2}-\frac{k_{1}}{B_{22}})u_{0}-\frac{k_{0}}{B_{22}}-\frac{k_{2}}{B_{22}}u_{0}{}^{2}-\frac{k_{3}}{B_{22}}u_{0}{}^{3}-\frac{k_{4}}{B_{22}}u_{0}{}^{4}-\frac{k_{5}}{B_{22}}u_{0}{}^{5}-\frac{k_{6}}{B_{22}}u_{0}{}^{6}\\ -\frac{k_{7}}{B_{22}}u_{0}{}^{7}+\frac{2B_{12}-2B_{21}}{B_{22}}u_{0}D_{0}^{2}u_{0}+\frac{4B_{11}-B_{20}-B_{02}}{B_{22}}u_{0}{}^{2}D_{0}^{2}u_{0}+\frac{2B_{10}-2B_{01}}{B_{22}}u_{0}{}^{3}D_{0}^{2}u_{0}-\frac{B_{00}}{B_{22}}u_{0}{}^{4}D_{0}^{2}u_{0}\\ +\frac{2\alpha_{1}\rho }{B_{22}}(v_{ac1}+v_{ac2}u_{0}+v_{ac2}u_{0}{}^{2})\cos(\omega t))\exp(-i\varphi )d\varphi \end{array}$$
where $\varphi =\omega T_{0}+\beta (T_{1})$.

Substituting Equation (37) into Equation (38) and separating the imaginary and real parts yields:(39)$$\left\{ \begin{array}{ll} \frac{da}{dT_{1}} & =\frac{ac^{\prime }C_{22}}{2B_{22}}-\frac{v_{ac1}\alpha_{1}\rho }{B_{22}\omega }\sin(\beta )-\frac{a^{2}v_{ac3}\alpha_{1}\rho }{4B_{22}\omega }\sin(\beta )\\ a\frac{d\beta }{dT_{1}} & =-\frac{ak_{1}}{2B_{22}\omega }+\frac{3a^{3}k_{3}}{8B_{22}\omega }+\frac{5a^{5}k_{5}}{16B_{22}\omega }+\frac{35a^{7}k_{7}}{128B_{22}\omega }-\frac{a\omega }{2}-\frac{3a^{3}B_{20}\omega }{8B_{22}}-\frac{3a^{3}B_{02}\omega }{8B_{22}}\\ & +\frac{3a^{3}B_{11}\omega }{2B_{22}}-\frac{5a^{5}B_{00}\omega }{16B_{22}}-\frac{v_{ac1}\alpha_{1}\rho }{B_{22}\omega }\cos(\beta )-\frac{3a^{2}v_{ac3}\alpha_{1}\rho }{4B_{22}\omega }cos(\beta ) \end{array}\right..$$

The steady-state response can be obtained by applying the condition $Da/DT_{1}=D\varphi /DT=0$. The ultimate frequency response can be deduced as follows:(40)$$\begin{array}{l} 4a^{2}c^{\prime }{}^{2}C_{22}^{2}\omega^{2}/{\left(4vac_{1}+a^{2}vac_{3}\right)}^{2}\alpha_{1}^{2}\rho^{2}+(64ak_{1}+48a^{3}k_{3}+40a^{5}k_{5}+35a^{7}k_{7}-48a^{3}B_{20}\omega^{2}\\ -48a^{3}B_{02}\omega^{2}+192a^{5}B_{11}\omega^{2}-40a^{5}B_{00}\omega^{2}-60aB_{22}\omega^{2}{)}^{2}/1024{\left(4vac_{1}+3a^{2}vac_{3}\right)}^{2}\alpha_{1}^{2}\rho^{2}=1 \end{array}$$

The micro-resonator vibration amplitude increases as the AC voltage amplitude increases. The physical parameters of the microbeam are L=400 μm, b=45 μm, d=3 μm, $h_{0}=2\;\mu m$, $V_{dc}=60\;V$, $E=1.65\times 10^{11}\;\mathrm{Pa}$, and $\rho =2.33\times 10^{3}\;\mathrm{kg}/m^{3}$. Select the upper section parameters $\lambda_{1}=-0.2$ and $\lambda_{2}=0.2$. The results from MMS, NMMS, and the Simulink dynamics simulation are compared in Figure 14, and the results from the three methods are found to be consistent when $V_{ac}=0.05\;V$. However, the MMS results begins to exhibit errors when $V_{ac}=0.1\;V$ as the vibration amplitude increases, while the NMMS results are still consistent with the simulation results. The system vibration appears to transition from hardening to softening when $V_{ac}=0.2V$. The new behavior can be obtained with NMMS. It is worth noting that the frequency response curve is composed of high and low energy branches when the vibration amplitude is small. As the amplitude increases, the two branches gradually converge. Since the main purpose of this chapter is to discuss the nonlinear softening-hardening transition behavior, it will not be discussed in further detail here. A detailed discussion can be found in literature [28].

Two kinds of motion can occur at the transition point: dynamic pull-in and jump to a higher stable branch. The energy output is higher in the second case. Changes in the softening-hardening transition point can be observed as the vibration amplitude increases from $V_{ac}=0.2\;V$ to $V_{ac}=0.25\;V$, $V_{ac}=0.3\;V$, and $V_{ac}=0.35\;V$, and the corresponding frequency response curve can be drawn. One can see from Figure 15 that the displacements corresponding to nonlinear softening-hardening transition points in the four cases are almost the same. It can be inferred that the position of the transition point does not change as the vibration amplitude increases or the change is negligible at the transition point.

According to previous studies, the location of the transition point is related to the microbeam thickness and DC voltage. The location of the transition point can be obtained by calculating du/dω=0, yielding the relationship between the thickness, DC voltage, and transition point. The transition point can only be obtained numerically because the resulting equation is more complicated. Curve fitting is used here to determine the transition point. There are two transition points in each frequency response curve. Figure 16 shows that the amplitudes corresponding to the two transition points are nearly equal. Therefore, only one curve can be used for the two positions. As can be seen from Figure 14, the position of softening-hardening transition point decreases with the increase of residual stress. Figure 16a shows the relationship between the thickness and the transition point position, and Figure 16b shows the relationship between the DC voltage and the transition point position. The straight line shows the solution drawn with the fitted equation, and the circle shows the numerical solution obtained by the Newton iteration. The green and blue circles show the left and right transition points, respectively. A comprehensive analysis of the two curves yields the following equation:(41)$$u=0.499-0.0057V_{dc}-1.1613(\lambda_{1}-\lambda_{2})-0.4846{\left(\lambda_{1}-\lambda_{2}\right)}^{2}.$$

Next, the relationship between the DC voltage, microbeam thickness, and transition point position is verified. Table 2 shows the case where the voltages are $V_{dc}=30\;V$, $V_{dc}=40\;V$, $V_{dc}=50\;V$, and $V_{dc}=60\;V$. Each case considers four different thicknesses to account for the breadth of the data. $u_{e}$ is the transition point position obtained from empirical equation, $u_{n\;l}$ and $u_{n\;r}$ are the positions of left and right transition points obtained by setting du/dω=0, respectively, $u_{n\;l}$ is the left transition point, and $u_{n\;r}$ is the right transition point. The “/” symbol indicates no transition point, i.e., there is no *softening-hardening* transition. Table 2 shows that the error between the transition point position and the actual transition point obtained from Equation (41) is small, which verifies the accuracy of the equation.

## 5. Finite Element Verification

### 5.1. Equivalent Natural Frequency Simulation

The microbeam resonator was explored and optimized using theoretical and numerical analysis. The finite element method will be used to simulate the optimization results using COMSOL Multiphysics software.

In COMSOL, the “solid mechanics” interface and “electrostatic” interface are combined with a dynamic grid function. First, the steady state of the model and the eigenfrequency in the steady state are studied. Here, finite element simulations are performed for the aforementioned seven cases, and the calculated simulation results are shown in Figure 17. One can see that the theoretical and simulation results are consistent, and the pull-in voltage at *A*, *B*, *C*, *D*, and *G* in Figure 17 are consistent with those in Figure 11.

The seven cases correspond to linear vibration when $V_{dc}=23.95\;V$. The equivalent theoretical frequency results are compared with the finite element equivalent frequency results when $V_{dc}=23.95\;V$ in Table 3. It is found that the maximum error is only 5.49%, which shows that the theoretical and numerical methods are consistent.

### 5.2. Equivalent Natural Frequency Simulation

The simulated equivalent natural frequency contains some error, although the error between the finite element solution and the theoretical solution is relatively small. More simulated frequency response curves are presented in this section. The physical parameters of the microbeam are L=400 μm, b=45 μm, d=3 μm, $h_{0}=2\;\mu m$, $V_{dc}=60\;V$, $E=1.65\times 10^{11}\;\mathrm{Pa}$, and $\rho =2.33\times 10^{3}\;\mathrm{kg}/m^{3}$. The simulated sectional parameters were $\lambda_{1}=-0.2$ and $\lambda_{2}=0.2$. Figure 16 shows a comparison between the simulation and theoretical results.

One can see that the amplitude response is linear when the amplitude is small, while hardening appears at higher amplitude. The red asterisk shows the results obtained from the frequency response analysis in COMSOL. One can see that the simulated amplitude is consistent with the theoretical amplitude when the AC voltage varies. When $V_{ac}=0.02V$, the simulated amplitude is 0.0409 and the theoretical amplitude is 0.0406, yielding a relative error of −0.73%. When $V_{ac}=0.1\;V$, the simulated amplitude is 0.204 and the theoretical amplitude is 0.203, yielding a relative error of 0.49%. However, because the frequency response analysis solver is a linear solver, the results do not exhibit hardening or softening. Each frequency is selected separately in order to further simulate the nonlinearity. After determining the frequency at a certain point, the corresponding amplitude is obtained with the transient solver. This step was repeated with different frequencies in order to determine the corresponding amplitude. This result is shown in blue circles in Figure 18. One can see that the finite element solution exhibits hardening. Moreover, the simulated natural frequency in the frequency domain is the same as the natural frequency in the time domain. The two simulation results are very consistent when the amplitude is small, but the maximum vibration amplitude from the time domain is lower than the other two results as the amplitude increases. This phenomenon may arise because the time interval corresponding to the highest point position is too short in the time domain analysis, thus the displacement does not point near the peak.

## 6. Conclusions

Static and dynamic analyses of electrostatically-driven microbeam resonators are presented in this paper. Two parametric equations describing control over the upper and lower sections of microbeam were introduced. The mechanical properties of the micro-resonator are strengthened by optimizing the parameters in these equations. The neutral plane tension, neutral plane bending, and electrostatic nonlinearity are considered in the mathematical model. First, the model is reduced and simplified by Galerkin discretization. Then the relationship between two section parameters under linear vibration is obtained by a Newton iteration. Finally, the vibration at small and large amplitudes were obtained using MMS and NMMS. The results were verified with the Runge-Kutta method, Simulink dynamic simulation and finite element method.

The key conclusions are as follows.The potential energy and electrostatic force are greater when the microbeam intermediate thickness is thinner for a given applied voltage. Decreasing the thickness at the midpoint of the microbeam will increase the pull-in voltage and increase the likelihood of secondary pull-in. The pull-in position is biased toward the larger of $\lambda_{1}$ and $-\lambda_{2}$ when microbeam is asymmetric about the neutral plane, and the pull-in voltage is primarily determined by the larger of $\lambda_{1}$ and $-\lambda_{2}$.When the micro-resonator vibrates linearly, the relationship between $\lambda_{1}$ and $\lambda_{2}$ is symmetric at y=x. The eigenfrequency and amplitude of the microbeam are studied when $\lambda_{1}$ and $\lambda_{2}$ correspond to linear vibration. Regarding the *y* = *x* symmetric point in Figure 7, the microbeams have the same eigenfrequency and different amplitudes. The amplitude near the AC voltage source is larger.The MMS should not be used as the amplitude increases. The frequency response obtained with MMS will exhibit weaker nonlinear softening. If the initial vibration exhibits hardening, the transition between hardening and softening will occur at a certain time as the vibration amplitude increases. The position of the transition point does not change as the amplitude continues to increase. At the transition point, the micro-resonator can jump to a higher stable branch and output more energy.

It is found that the static and dynamic behaviors of micro-resonator can be changed by adjusting the microbeam shape. Determining the appropriate geometry can improve the mechanical properties for given raw materials.

## Figures and Tables

**Figure 1 sensors-19-01348-f001:**
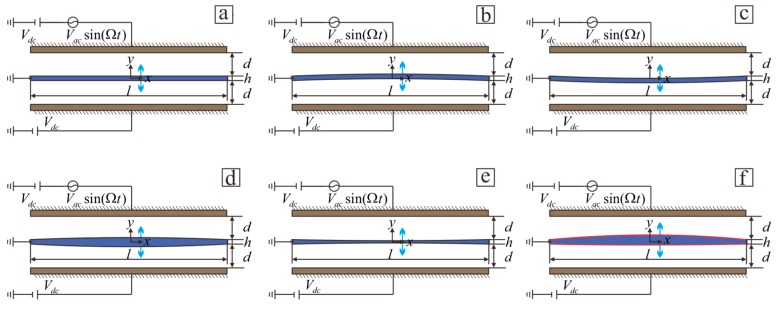
Schematic of an electrically actuated microbeam: (**a**) rectangular beam; (**b**) microbridge bending upward; (**c**) microbridge bending downward; (**d**) thickened microbeam; (**e**) thinned microbeam; (**f**) irregular beam.

**Figure 2 sensors-19-01348-f002:**
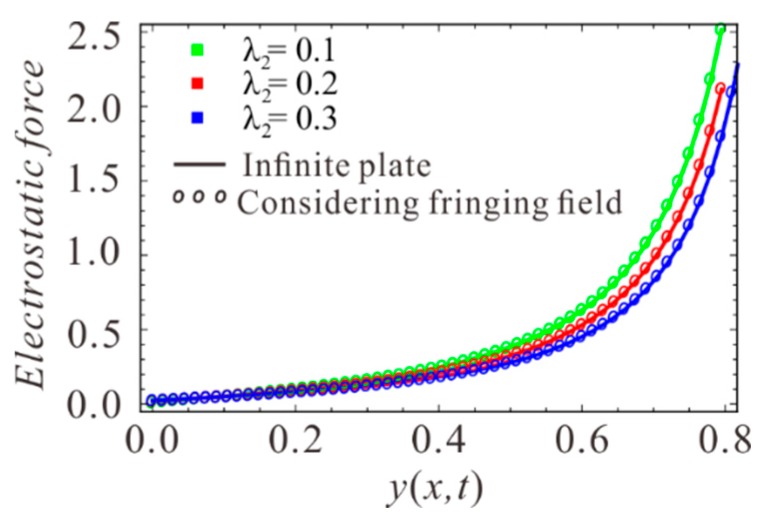
The comparisons of electrostatic forces under different $\lambda_{2}$ at $\lambda_{1}=0$. Solid lines show solution of the infinite plate model and circle show solution considering the fringing fields.

**Figure 3 sensors-19-01348-f003:**
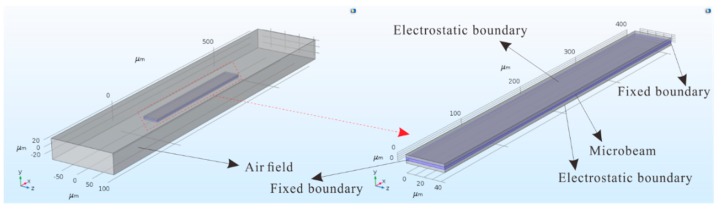
COMSOL simulation model.

**Figure 4 sensors-19-01348-f004:**
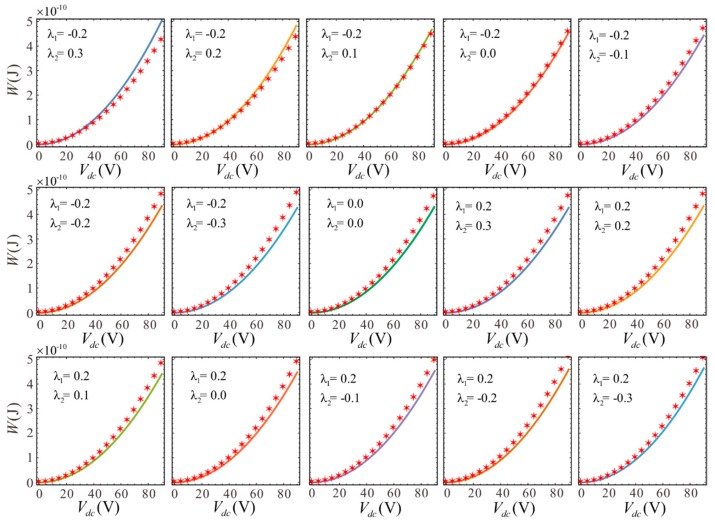
The relationship between the DC voltage and potential energy for different values of section parameters. Lines show the analytical solutions and symbols show the finite element solutions.

**Figure 5 sensors-19-01348-f005:**
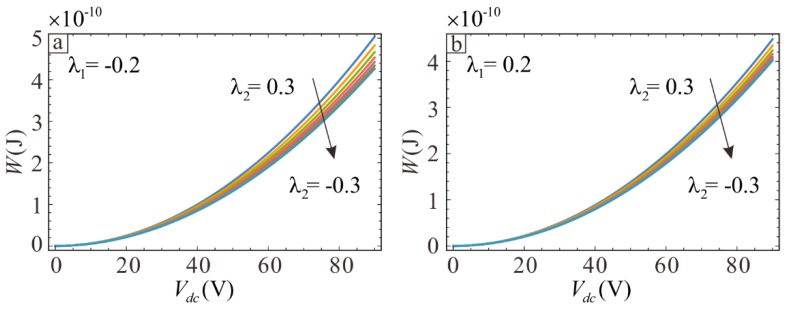
The relationship between the DC voltage and potential energy for different values of section parameters. (**a**) the case of $\lambda_{1}=-0.2$; (**b**) the case of $\lambda_{1}=0.2$.

**Figure 6 sensors-19-01348-f006:**
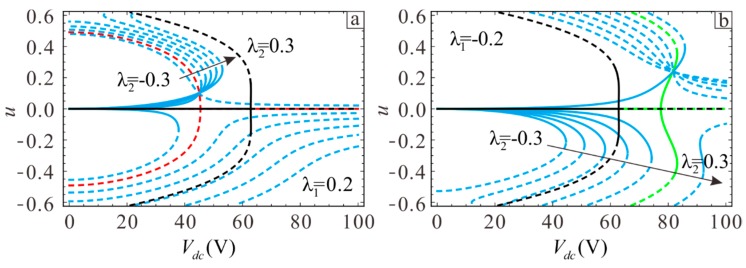
Influence of the section parameters on static pull-in: (**a**) $\lambda_{1}=0.2$ and (**b**) $\lambda_{1}=-0.2$. Solid lines show stable solutions and dashed lines show unstable solutions. Blue lines indicate asymmetry, while red and green lines indicate symmetry. The black line indicates the case where the section parameters are $\lambda_{1}=\lambda_{2}=0$.

**Figure 7 sensors-19-01348-f007:**
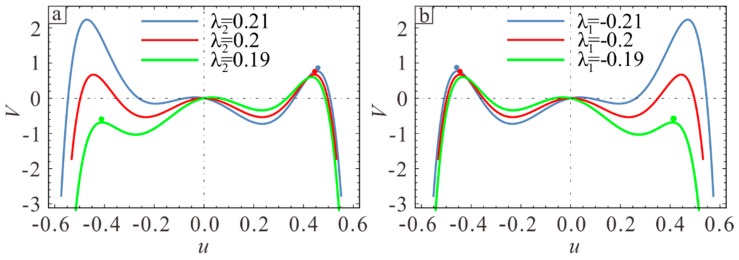
The influence of section parameters on potential energy. (**a**) The influence of $\lambda_{2}$ on static pull-in when $\lambda_{1}=-0.2$. (**b**) The influence of $\lambda_{1}$ on static pull-in when $\lambda_{2}=0.2$.

**Figure 8 sensors-19-01348-f008:**
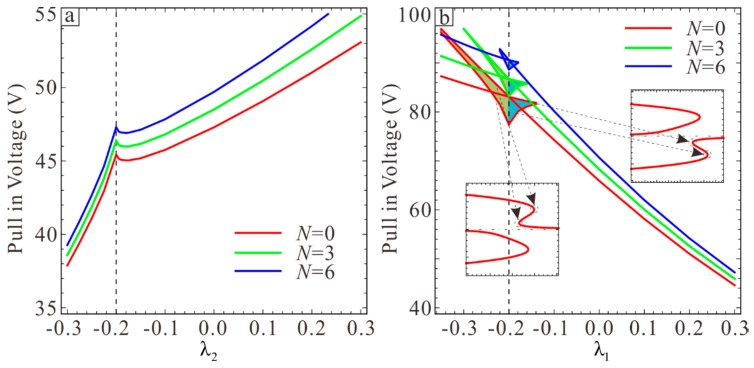
The influence of section parameters on pull-in voltage under different residual stresses. (**a**) The influence of $\lambda_{2}$ on pull-in voltage when $\lambda_{1}=0.2$. (**b**) The influence of $\lambda_{1}$ on pull-in voltage when $\lambda_{2}=0.2$.

**Figure 9 sensors-19-01348-f009:**
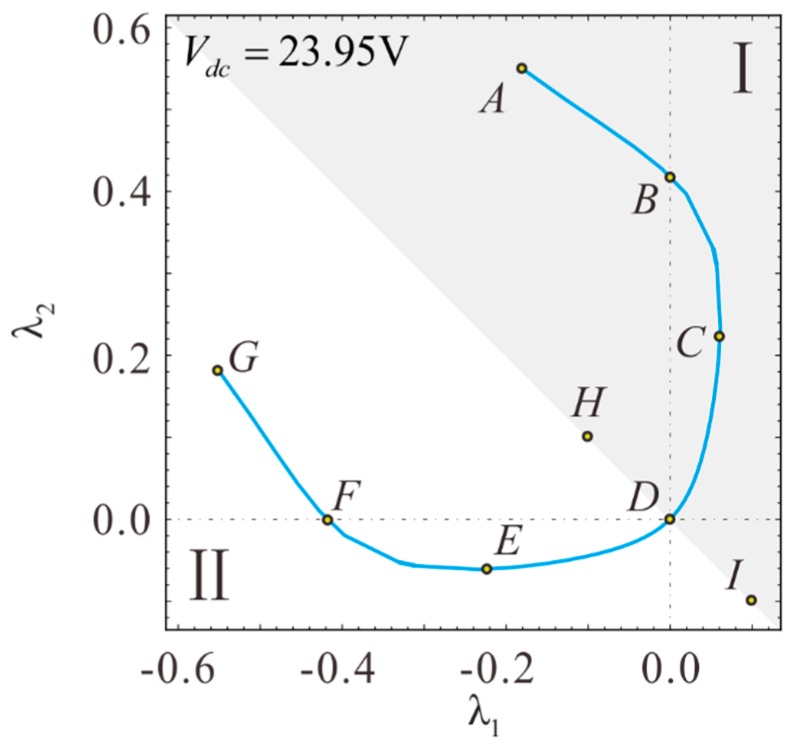
The relationship between $\lambda_{1}$ and $\lambda_{2}$ under linear vibration when $V_{dc}=23.95\;V$.

**Figure 10 sensors-19-01348-f010:**
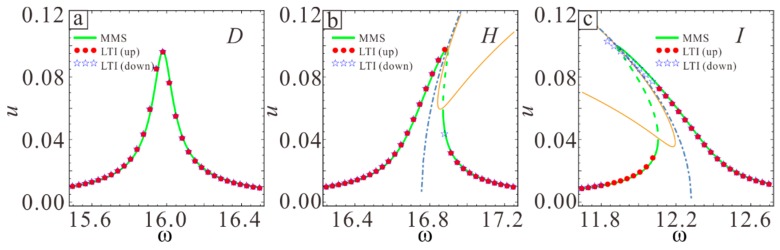
Frequency response curves for D (**a**), H (**b**), and I (**c**). The solid line shows the theoretically stable solution and the dotted line shows the theoretically unstable solution.

**Figure 11 sensors-19-01348-f011:**
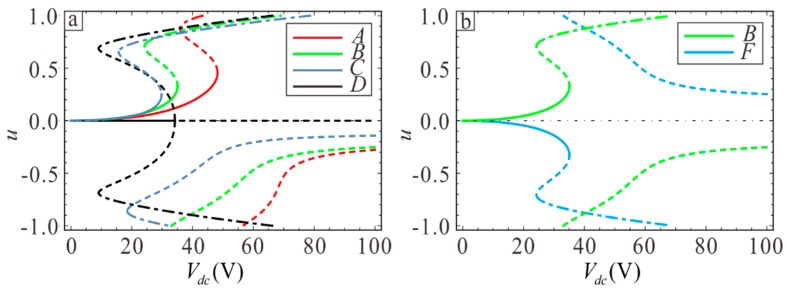
The relationship between the DC voltage and the static equilibrium point. (**a**) Static equilibrium point for A, B, C, and D. (**b**) Static equilibrium point for B and F cases. Solid lines represent stable solutions and dashed lines represent unstable solutions. The dotted line shows the stable solution without physical meaning.

**Figure 12 sensors-19-01348-f012:**
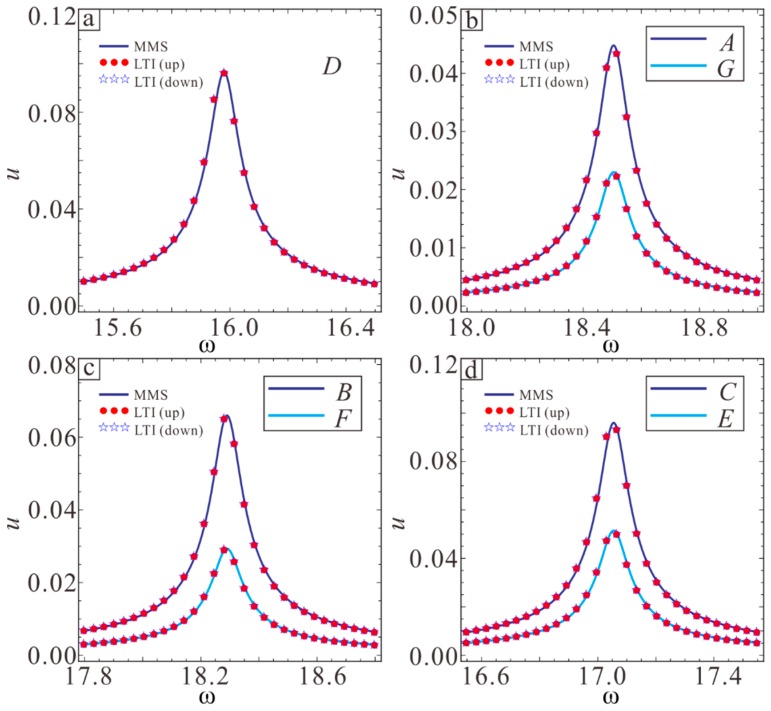
Frequency response curves for *A*, *B*, *C*, *D*, *E* and *F* cases: (**a**) the case of *D*; (**b**) the cases of *A* and *G*; (**c**) the cases of *B* and *G*; (**d**) the cases of *C* and *E*.

**Figure 13 sensors-19-01348-f013:**
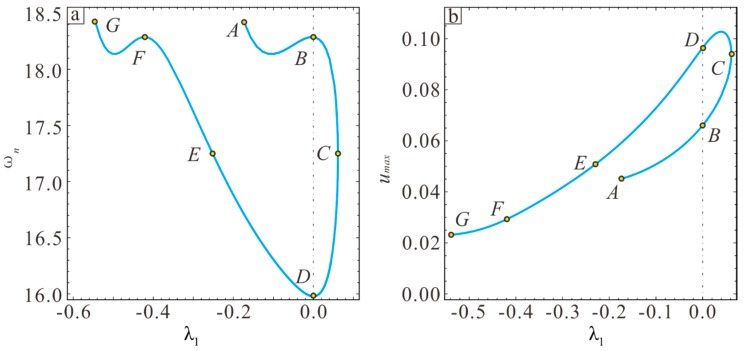
Influence of the section parameters on system vibration characteristics under linear vibration. (**a**) Influence of the cross-section parameters on the equivalent natural frequencies under linear vibration. (**b**) Influence of the section parameters on vibration amplitude under linear vibration.

**Figure 14 sensors-19-01348-f014:**
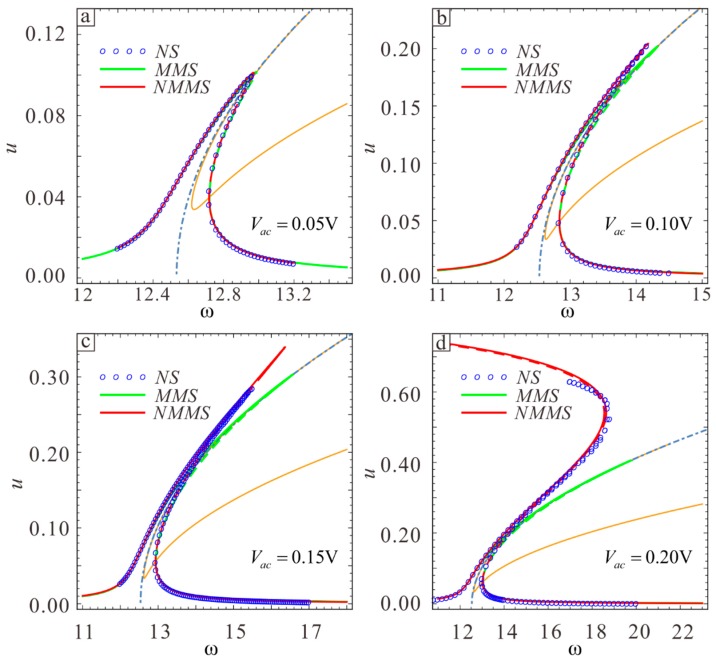
Influence of the AC voltage on the frequency response curve when (**a**) $V_{ac}=0.05\;V$, (**b**) $V_{ac}=0.1\;V$, (**c**) $V_{ac}=0.15\;V$, and (**d**) $V_{ac}=0.20\;V$. The green straight line shows the theoretically stable solution obtained with MMS. The green dotted line shows the theoretically unstable solution obtained with MMS. The red straight line shows a theoretically stable solution obtained with NMMS. The red dotted line shows the theoretically unstable solution obtained with NMMS. Circles show the numerical solution obtained with the Simulink dynamics simulation (NS).

**Figure 15 sensors-19-01348-f015:**
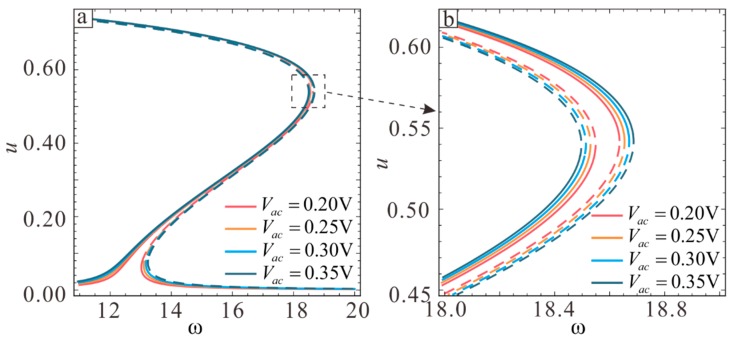
Influence of the AC voltage on the frequency response curve. The straight line shows the stable solution and the dotted line shows the unstable solution.

**Figure 16 sensors-19-01348-f016:**
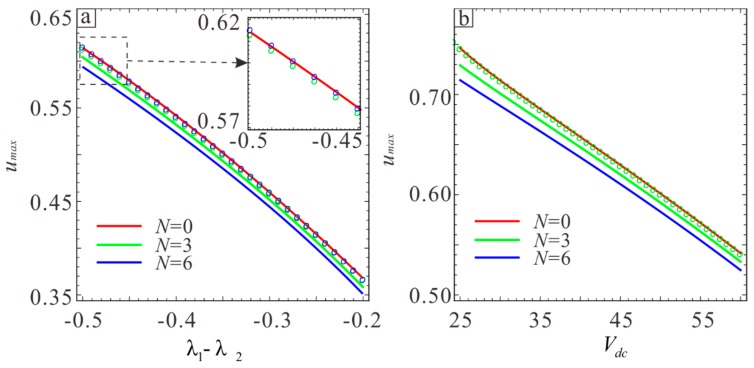
The relationship between the transition points and physical parameters of the system under different residual stresses. (**a**) The relationship between the transition points and $\lambda_{1}-\lambda_{2}$. (**b**) The relationship between the transition points and DC voltage. The straight line shows the result obtained with Equation (41). Circles show the exact position of the transition point obtained by the Equation (40).

**Figure 17 sensors-19-01348-f017:**
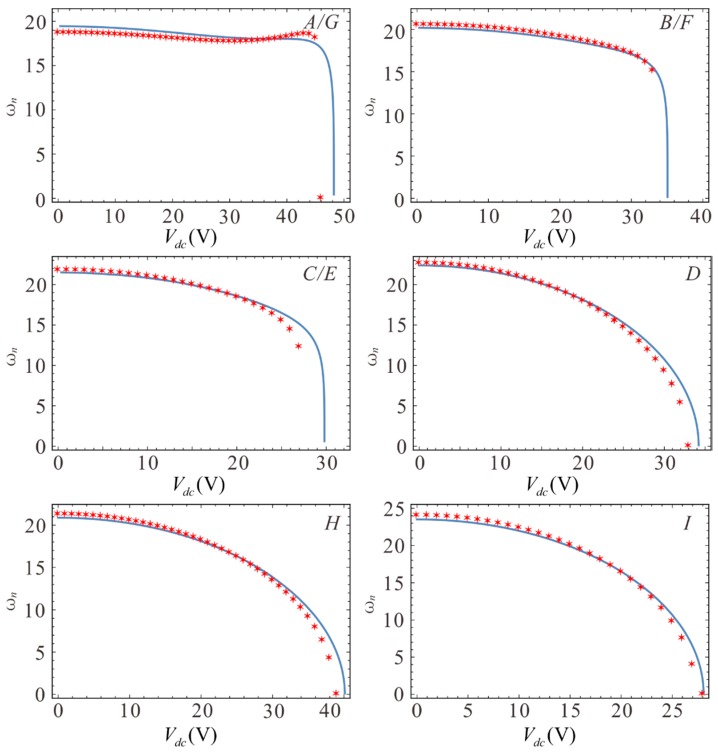
Electrostatic force softening effect in different cases. Lines show the analytical solutions and pentagons show the finite element solutions.

**Figure 18 sensors-19-01348-f018:**
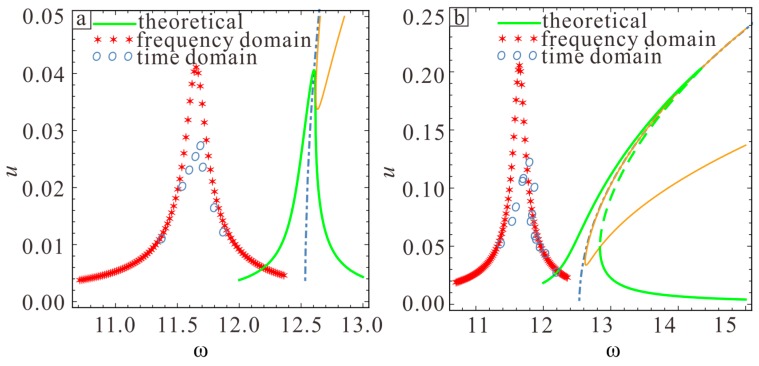
Comparison of frequency response curves when (**a**) $V_{ac}=0.02\;V$ and (**b**) $V_{ac}=0.1\;V$. The line shows the analytical solution, asterisks show the finite element solution obtained with the frequency domain solver, and circles show the finite element solution obtained with the time domain solver.

**Table 1 sensors-19-01348-t001:** Section parameter corresponding to nine special points.

Special Points	$\lambda_{1}$	$\lambda_{2}$
*A*	−0.18	0.55
*B*	0	0.419
*C*	0.061	0.225
*D*	0	0
*E*	−0.225	−0.061
*F*	−0.419	0
*G*	−0.55	0.18
*H*	−0.1	0.1
*I*	0.1	−0.1

**Table 2 sensors-19-01348-t002:** The comparison of transition points obtained with two methods.

$V_{dc}\left(V\right)$	$\lambda_{1}-\lambda_{2}$	$u_{e}$	$u_{n\;l}$	$u_{n\;r}$	Error
60	−0.6	0.6793	0.6882	0.6905	−1.29%/−1.62%
	−0.5	0.6165	0.6133	0.6156	0.52%/0.15%
	−0.4	0.5440	0.5306	0.5326	2.53%/2.14%
	−0.3	0.4618	0.4364	0.4373	5.82%/5.60%
50	−0.5	0.6735	0.6729	0.6757	0.09%/−0.33%
	−0.4	0.6010	0.5927	0.5958	1.40%/0.87%
	−0.3	0.5188	0.5045	0.5070	2.83%/2.33%
	−0.2	0.4269	0.4046	0.4049	5.51%/5.43%
40	−0.4	0.6580	0.6546	0.6585	0.52%/−0.08%
	−0.3	0.5758	0.5686	0.5722	1.27%/0.63%
	−0.2	0.4839	0.4742	0.4767	2.05%/1.51%
	−0.1	0.3823	0.3682	0.3674	3.83%/4.06%
30	−0.3	0.6328	0.6349	0.6412	−0.33%/−1.31%
	−0.2	0.5409	0.5386	0.5439	0.43%/1.12%
	−0.1	0.4393	0.4370	0.4390	0.53%/0.07%
	0.0	0.3280	0.3254	0.3211	0.80%/2.15%

**Table 3 sensors-19-01348-t003:** Comparison of the finite element simulation and theoretical results.

	*A*/*G*	*B*/*F*	C/E	*D*	*H*	*I*
Theoretical frequency	18.507	18.289	17.050	15.973	16.752	12.269
Simulation frequency	17.876	18.530	16.371	15.354	16.621	11.631
Error	3.51%	−1.30%	4.15%	4.03%	0.79%	5.49%

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
