# Peer review of "Mechanical Behaviors Research and the Structural Design of a Bipolar Electrostatic Actuation Microbeam Resonator"

_sensors, 2019, doi:10.3390/s19061348_

Reviewer 1 Report

This work presents an interesting study about the impact of non-symmetrical cc-beam shape on the overall system behavior. While the work is interesting there are some aspects that should be further analyzed before it is suitable for publication. The most significant are:

Authors do not consider the impact of fringing fields on the overall system behavior. Previous works have shown that the effect of fringing fields on the final pull-in effect is significant. Looking at your results such an effect could be the cause of your model deviation at large voltages. I suggest including a discussion about this in the paper if accepted.

While authors mention the effect of the residual stress its value is not quantified and the impact of it on the final results should also be discussed in more detail.

Your equation 13 obtained after the Galerkin method represents an equivalent mass-spring system. Such an approximation neglects the specific deflection shape of the cc-beam along the beam length. The validity of this approximation should be justified more rigorously.

Looking at your results in Fig 4. it seems that the impact of the lambda parameters variation on the final potential is not so significant, this is further confirmed in Fig 6, where it is shown that to have asymmetry in the potential distribution the deflection must get a significant displacement that is clearly out of the linear regime. Could you please comment on this?

Why are you considering sinusoidal variations of the beam thickness? would other shapes be more appropriated for you objectives? please comment.

It would be also interesting to discuss how practical would be fabricating these structures, specially when considering process parameter variations. Is this structure really doable?

Could you please specify in more detail what does it really mean to consider that the volume of the microbeam does not change, when the shape does change?

Finally a minor comment on Fig. 1: consider drawing the dc and ac voltage sources in series instead of short-circuiting them.

Author Response

Review Report 1

Thank you for your suggestions and questions. We have carefully answered your questions and revised them in the manuscript. The specific answers are as follows

1.

Question:

Authors do not consider the impact of fringing fields on the overall system behavior. Previous works have shown that the effect of fringing fields on the final pull-in effect is significant. Looking at your results such an effect could be the cause of your model deviation at large voltages. I suggest including a discussion about this in the paper if accepted.

Reply:

Thank you for your question. As is known to all, when the ratio of gap distance to microbeam width is small, the fringing fields of electrostatic force is obvious and it cannot be ignored at this time. But as this ratio increases, the impact of fringing fields gradually decreases. In our manuscript, the ratio of gap distance to microbeam width is 22.5, the errors of electrostatic force considering the fringing fields and electrostatic force without considering t fringing fields are almost negligible. We have added the contrast picture of electrostatic force in our revised manuscript and further explained them. Details can be found in line 176-182.

2.

Question:

While authors mention the effect of the residual stress its value is not quantified and the impact of it on the final results should also be discussed in more detail.

Reply:

We have added the influence of residual stress on the pull-in voltage, secondary pull-in phenomenon and softening-hardening transition point in our revised manuscript. Details can be found in line 292-308; 474-478; 484-487.

3.

Question:

Your equation 13 obtained after the Galerkin method represents an equivalent mass-spring system. Such an approximation neglects the specific deflection shape of the cc-beam along the beam length. The validity of this approximation should be justified more rigorously.

Reply:

Thank you for your question. In our manuscript, the original fourth-order partial differential equation is transformed into second-order ordinary differential equation by Galerkin method and its applicability has been confirmed in many previous articles, such as Reduced-Order Model for Electrically Actuated Microbeam-Based MEMS. Equation 13 is the Hamiltonian form of equation 11, which is equivalent to equation 11. Parameters                                                in equation 13 is an expression related to deflection displacement、lambda 1 and lambda 2. It accurately contains the information of CC beam shape variety and deflection displacement. 

4.

Question:

Looking at your results in Fig 4. it seems that the impact of the lambda parameters variation on the final potential is not so significant, this is further confirmed in Fig 6, where it is shown that to have asymmetry in the potential distribution the deflection must get a significant displacement that is clearly out of the linear regime. Could you please comment on this?

Reply:

Thank you for your question. Fig. 6 studies the influence of lambda 1 and lambda 2 on the system pull-in. The physical parameters of microbeam selected in Fig. 6 satisfy the secondary pull-in phenomenon. The microbeam will displace under the action of electrostatic force. With the increase of electrostatic force, the displacement increases gradually and finally the microbeam will be pull-in together with the electrode plate. The obvious displacement you mentioned is the displacement of microbeam under the action of electrostatic force.

5.

Question:

Why are you considering sinusoidal variations of the beam thickness? would other shapes be more appropriated for you objectives? please comment.

Reply:

Thank you for your question. Our manuscript mainly discusses the influence of microbeam thickness variation on the mechanical properties of the system and optimizing the model by adjusting the control parameters. Here the research is only a preliminary assumption of a simple shape change function, which is used to verify whether the change of thickness will have a great influence on the mechanical behavior of micro-resonator. We can observe influence trend of microbeam shape on the resonator mechanical properties through the section change function. For other functions, this method is also applicable as long as the selected function satisfies the model.

6.

Question:

It would be also interesting to discuss how practical would be fabricating these structures, specially when considering process parameter variations. Is this structure really doable?

Reply:

  Thank you for your question. Nowadays, the fabrication process of MEMS has been able to fabricate complex three-dimensional models through photolithography, epitaxy, film deposition, oxidation, injection, sputtering, evaporation, etching, scratching and packaging. The devices fabricated can also reach nanometer scale. The microbeam model is relatively simple, so it can be processed with the present technology. However, because the beam thickness is sinusoidal variations, there may be errors in processing accuracy. The microbeam shape variation caused by errors is random, so it is impossible to accurately calculate the influence of errors. This manuscript was research based on ideal processing and the main purpose is to observe the influence of thickness variation on mechanical behaviors of system. We need future in-depth research to get close to the actual situatin.

7.

Question:

Could you please specify in more detail what does it really mean to consider that the volume of the microbeam does not change, when the shape does change?

Reply:

Thank you for your question. The constant-volume constraint helps in comparing the two designs (rectangle and non-rectangular), for their performance, when both of them consume same amount of material during their fabrication. This constraint also precludes the role of the volume of an actuator in controlling its pull-in parameters, and thus allows us to focus on the actuator’s shape instead. Detailed explanations are added to line 142-144 of the manuscript.

8.

Question:

Finally a minor comment on Fig. 1: consider drawing the dc and ac voltage sources in series instead of short-circuiting them.

Reply:

  Thank you for your suggestion. We have modified the dc and ac voltage sources connection method in the revised manuscript.

Reviewer 2 Report

Comments to the Authors:

This manuscript deals with vibrations of an electrostatically actuated microbeam resonator. The equation of motion is derived considering electrostatic force nonlinearity, neutral surface tension, and neutral surface bending effects. The approach and methods are standard. I have the following specific comments, which the authors should address before publication:

1) The title is not clear. Please revise it.

2) Abstract should be concise and clear.

3) Please mention the sources of nonlinearity in the equation of motion. Do you have the geometric nonlinearity?

4) Have you considered the effect of intermolecular interactions such as Van der Waals or Casimir?

5) Figure 16 needs a legend.

6) What is the value of damping in your analytical method and the finite element simulation? Why have you considered this value?

7) New class of resonators such as disk [1, 2] and ring resonators [3, 4]  are getting a lot of attention. Please provide this to the readers and mention it in the introduction section.

8) Which of the methods you have used can predict the size-dependency?

[1] Chorsi, M. T., and Chorsi, H. T., 2018, "Modeling and analysis of MEMS disk resonators," Microsystem Technologies, 24(6), pp. 2517-2528.

[2] Chorsi, H. T., Chorsi, M. T., and Gedney, S. D., 2017, "A conceptual study of microelectromechanical disk resonators," IEEE Journal on Multiscale and Multiphysics Computational Techniques, 2, pp. 29-37.

[3] Xie, Y., Li, S.-S., Lin, Y.-W., Ren, Z., and Nguyen, C. T.-C., 2008, "1.52-GHz micromechanical extensional wine-glass mode ring resonators," ieee transactions on ultrasonics, ferroelectrics, and frequency control, 55(4), pp. 890-907.

[4] Chorsi, M. T., Chorsi, H. T., and Gedney, S. D., 2017, "Radial-contour mode microring resonators: Nonlinear dynamics," International Journal of Mechanical Sciences, 130, pp. 258-266.

Author Response

Review Report 2

Thank you for your suggestions and questions. We have carefully answered your questions and revised them in the manuscript. The specific answers are as follows

1.

Question:

The title is not clear. Please revise it.

Reply:

Thank you for your suggestion. We have modified the title in revised manuscript.

2.

Question:

Abstract should be concise and clear.

Reply:

We have modified the abstract in revised manuscript.

3.

Question:

Please mention the sources of nonlinearity in the equation of motion. Do you have the geometric nonlinearity?

Reply:

The nonlinear in the motion equation is mainly includes the neutral surface tension, neutral surface bending caused by thickness variation and electrostatic force nonlinear, where the neutral surface tension and bending belong to geometric nonlinearity.

4.

Question:

Have you considered the effect of intermolecular interactions such as Van der Waals or Casimir?

Reply:

Because intermolecular forces such as Van der Waals or Casimir are only significant when the model size is nanoscale. In our manuscript, the model maximum size is 400 microns and the minimum size is 2 microns. The influence of intermolecular force is far less than that of electrostatic force and its influence on the system can be neglected. Therefore, the intermolecular forces are not considered in our manuscript.

5.

Question:

Figure 16 needs a legend.

Reply:

The legend has been added. After modification, Figure 16 in the manuscript becomes Figure 18. Details can be found in line 532-535.

6.

Question:

What is the value of damping in your analytical method and the finite element simulation? Why have you considered this value?

Reply:

In our manuscript, the dimensionless damping coefficient selected in theoretical analysis is 0.1 and the original dimensionless damping coefficient is about 0.000639332 N/m/s. In the finite element software, Rayleigh damping is set, the mass damping parameter is 68.1726 [1/s] and the stiffness damping parameter is 1.03223e-8 [s]. The specific values of the damping vary with the parameters lambda1 and lambda2, but the variation range is small and can be neglected. Because the microbeam model in our manuscript is a polycrystalline silicon material, the damping coefficient of silicon is between 0.0006 and 0.02. First, the theoretically selected damping coefficient should be within this range. Then in order to ensure that the dimensionless maximum amplitude at resonance is close to 0.1 when DC voltage and AC voltage is applied. Therefore, the dimensionless damping coefficient chosen in this paper is 0.1.

7.

Question:

New class of resonators such as disk [1, 2] and ring resonators [3, 4] are getting a lot of attention. Please provide this to the readers and mention it in the introduction section.

Reply:

Thank you for your suggestion. We have added these literatures in revised manuscript.

8.

Question:

Which of the methods you have used can predict the size-dependency?

Reply:

Thank you for your question. The purpose of this paper is to study the influence of microbeam thickness variation on the mechanical properties of system and to optimize the structure of microbeam model. In order to convenient for analysis, the size effect is not considered. If we want to predict the structure size effect, the strain gradient elasticity theory is a feasible method. This theory considers the effects of three strain gradients (the symmetric component of the rotational gradient tensor, the skewed component of the expansion gradient tensor and the skewed component of the tensile gradient tensor) on the strain energy density and introduces intrinsic parameters related to the characteristic size of material microstructures in the constitutive relationship, which provides a theoretical basis for simulating the size effect phenomenon.

Round  2

Reviewer 1 Report

Authors have addressed the concerns raised in the first review, therefore from our side it can be published.

Reviewer 2 Report

The authors have addressed all my comments. I suggest its publication in Sensors.